# Efficacy and Sustainability of Diabetes-Specific Meal Replacement on Obese and Overweight Type-2 Diabetes Mellitus Patients: Study Approaches for a Randomised Controlled Trial and Impact of COVID-19 on Trial Progress

**DOI:** 10.3390/ijerph19074188

**Published:** 2022-04-01

**Authors:** Leong Chen Lew, Arimi Fitri Mat Ludin, Suzana Shahar, Zahara Abdul Manaf, Noorlaili Mohd Tohit

**Affiliations:** 1Biomedical Science Programme, Universiti Kebangsaan Malaysia, Jalan Raja Muda Abdul Aziz, Kuala Lumpur 50300, Malaysia; p104558@siswa.ukm.edu.my; 2Center for Healthy Ageing and Wellness, Universiti Kebangsaan Malaysia, Jalan Raja Muda Abdul Aziz, Kuala Lumpur 50300, Malaysia; suzana.shahar@ukm.edu.my (S.S.); zaharamanaf@ukm.edu.my (Z.A.M.); 3Dietetic Programme, Universiti Kebangsaan Malaysia, Jalan Raja Muda Abdul Aziz, Kuala Lumpur 50300, Malaysia; 4Department of Family Medicine, University Kebangsaan Malaysia Medical Centre (UKMMC), Cheras, Kuala Lumpur 56000, Malaysia; laili@ppukm.ukm.edu.my

**Keywords:** diabetes, obesity, diet modification, meal replacement, glycemic control, HbA1c

## Abstract

Meal replacement (MR) is widely used in weight and diabetes management programs due to its ease of compliance and handling. However, little is known about its impact on outcomes other than glycaemic control and weight loss. Furthermore, not many studies evaluate its cost-effectiveness and sustainability. This study aimed to evaluate the efficacy of a diabetes-specific MR for the weight reduction and glycaemic controls of overweight and obese T2DM patients, as compared to routine dietary consultation. Other health outcomes, the cost effectiveness, and the sustainability of the MR will also be evaluated. Materials and Methods: This randomised controlled clinical trial will involve 156 participants who have been randomised equally into the intervention and control groups. As a baseline, both groups will receive diet consultation. Additionally, the intervention group will receive an MR to replace one meal for 5 days a week. The duration of intervention will be 12 weeks, with 36 weeks of follow-up to monitor the sustainability of the MR. The primary endpoints are weight and Hemoglobin A1c (HbA1c) reduction, while the secondary endpoints are anthropometry, biochemical measurements, satiety, hormone changes, quality of life, and cost-effectiveness. The impact of the COVID-19 pandemic on study design is also discussed in this paper. This study has obtained human ethics approval from RECUKM (JEP-2019-566) and is registered at the Thai Clinical Trials Registry (TCTR ID: TCTR20210921004).

## 1. Introduction

Obesity is one of the biggest ongoing health concerns, with 1.9 billion adults worldwide who are either overweight or obese [1]. Among Southeast Asia countries, Malaysia has the highest prevalence of obesity among adults [2]. Based on the National Health and Morbidity Survey 2019 findings, the prevalence of an overweight or obese status among Malaysian adults has reached a concerning level of 50.1% [3].

Obesity is closely linked to Type-2 Diabetes Mellitus (T2DM). In fact, obesity itself causes insulin resistance, specifically in visceral obesity. In 38 out of 44 studies pooled in a systematic review on the prevalence of obesity among T2DM patients, over 30% of participants are reported as obese [4].

As reported by the International Diabetes Foundation (2017), approximately 451 million people worldwide are affected by T2DM, and the number is expected to increase to 693 million by the year 2045 [5,6]. The latest National Health and Morbidity survey has reported that the prevalence of diabetes in Malaysia is 18.3%, with an increase of 4.9% since 2015 [3].

Efforts for effective lifestyle intervention and management of T2DM tailored for overweight and obese patients are essential for weight reduction and the improvement of glycaemic control among T2DM patients. Meal replacement is a term which refers to pre-packaged or commercially available food products or drinks used to replace one or more meals [7]. A systematic review that involves 23 studies and 7884 adults showed that subjects who received meal replacement lost more weight, compared to those who received other diet plans or only consultation [8]. Furthermore, incorporating meal replacement as part of a comprehensive lifestyle intervention has also been found to be effective in reducing HbA1C and facilitating the initial weight loss of 5145 overweight or obese T2DM participants, as reported in one of the largest diabetes weight management studies, the Look AHEAD study [9,10,11].

However, there are still limitations faced by some of the existing meal replacement formulations, such as the risks of side effects. There is a cohort study on 8361 participants that shows that a very low-calorie diet (VLCD) increases the risks of symptomatic gallstones that require a cholecystectomy [12]. Certain liquid meal replacements do not contain enough fibre, which may lead to undesired effects in the bowel system if consumed daily. T2DM patients, who require daily adequate nutrient intake, also often suffer from nutrient deficiencies [13].

While meal replacement can lead to greater initial weight loss and glucose control, there is still a lack of evidence on the long-term feasibility of the sustained use of meal replacements [14]. Once the meal replacement intervention is reduced or discontinued, its benefit of glycemic control and weight reduction may not be sustained [15]. The cost of meal replacement might also be a barrier for long-term consumption, as it can be an economic burden for patients. In most clinical trials, meal replacement is often subsidized, or provided for free, to subjects during the initial intervention. Upon completing the initial intervention, participants are often unwilling to pay for long term therapy [16]. Once the subsidised meal replacement is reduced or completely discontinued, the benefits may be reversed, as shown in a study by Keogh and Clifton [15].

Hence, in this study we are going to evaluate the effectiveness and sustainability of a nutritionally balanced diabetes-specific meal replacement in weight reduction and glycaemic parameters controlled for overweight and obese patients with T2DM, as compared to normal dietary consultation. We are also going to assess the effect of meal replacement on satiety level and hormonal changes, as well as the cost-effectiveness of meal replacement when compared to standard care. This will be the first randomised controlled trial to utilise the diabetes-specific meal replacement *Metabolic Sauver* among T2DM patients.

The results from this study can help in providing input towards the future development of lifestyle modification modules that incorporate short-term meal replacement therapy to be utilised in clinical settings. This can help in our fight against the disease and ease the burden of T2DM on the nation’s healthcare and economy.

In this article, we also discussed the impact of the COVID-19 pandemic on the recruitment and enrollment process of this ongoing clinical trial. The COVID-19 pandemic has affected over 172 million people worldwide, with 616,845 people testing positive for COVID-19 in Malaysia as of June 2021 [17]. It remains a crisis in the country as authorities have redirected most of their resources towards handling COVID-19 patients. The measures taken by the authorities can be impactful towards clinical trial research, which is deemed as a desirable but not an essential activity. The pandemic can affect clinical trials in various ways, such as the completion of trial assessments, patient visits to clinical settings, and stock shortages for investigational medicinal products. It is important to assess and understand the impact of the COVID-19 pandemic for effective countermeasures to ensure the success of our clinical trial.

### 1.1. Research Objective

Our research objective is to determine the effect and sustainability of diabetes-specific meal replacement on weight reduction (weight, BMI, body composition) and glycemic control (HbA1c, fasting blood glucose) among obese or overweight T2DM participants. We are also going to assess the effect of meal replacement on satiety level, hormonal changes (adiponectin, leptin, ghrelin, obestatin, peptide YY) and the quality of life among the participants. Cost-effectiveness will also be evaluated, as compared to standard care.

#### 1.1.1. Primary Endpoint

Changes in HbA1c and bodyweight from the beginning of treatment to week 12 after initiation of treatment

#### 1.1.2. Secondary Endpoint

BMIBody CompositionWaist and hip circumferenceFasting blood glucoseInsulin resistance index (HOMA-IR)Lipid ProfileSatiety levelHormonal ChangesQuality of lifeCost-effectiveness

## 2. Methodology

### 2.1. Study Design

The study design is a two-armed randomised controlled clinical trial. Participants will be recruited into the study based on the inclusion and exclusion criteria (Table 1). Five thousand one hundred and eighty-five patients are screened through digital medical records in the database for their eligibility to be potentially recruited into the study.

Eligible and consenting participants will be randomised into two groups: (a) Meal replacement group (MR) and (b) Control group. The MR group will be receiving a meal replacement product and dietary consultation, while the control group will be receiving dietary consultation only. Outcome measurements will be assessed at the baseline, 6th week and after the completion of the 12-week intervention. Sociodemographic data, dietary information, and the quality of life of participants will also be recorded via questionnaires. Participants will also be monitored for 36 weeks after the completion of the meal replacement without any additional intervention. Another follow up will be done in week 48 and the same parameters will be measured. Hence, the total duration of the study is 48 weeks. A brief research flow chart is shown below in Figure 1.

### 2.2. Sample Size

Sample size is calculated using G *Power 3.1 software. The protocol for calculation is as shown below in Table 2. We employed an effect size of 0.2055067, which is obtained from the mean difference value of the HbA1c (%) result in a study by Keogh and Clifton [18]. Based on the calculation, a total of 120 participants is needed for the study. In order to account for up to a 30% dropout, a total of 156 participants will be recruited for this study.

### 2.3. Recruitment

Participants will be recruited from the outpatient clinic of Hospital Canselor Tuanku Muhriz’s Primer Clinic of the National University of Malaysia via purposive sampling. Potential participants will be identified from the computerized medical record database according to the inclusion and exclusion criteria stated in Table 1. Invitations will be sent to potential participants via phone call and email. Interested participants will be invited to attend a screening. Prior to the screening process, detailed information about this study will be explained, and written consent (Online Appendix A) will be obtained. Recruitment will be an ongoing process throughout the study until the required number of participants is achieved. This study will be carried out in accordance with the Helsinki Declaration of 1975, as revised in 1989, which states that no vulnerable participants should be recruited. The study is currently in the recruitment phase. The screening process started in September 2020. The recruitment process has been halted a few times due to the COVID-19 pandemic in Malaysia.

### 2.4. Baseline Screening and Assessment

Subsequent to the recruitment, participants who have provided consent will be directed to the clinic for baseline assessment. Anthropometry and body composition measurements will be recorded. A research dietician will assess the dietary intake of the participants with a 3-day dietary recall and food frequency questionnaire on added sugar intake. Eighteen millilitres of blood will be collected by a qualified phlebotomist to measure HbA1c, fasting plasma glucose, fasting insulin, lipid, renal, liver profiles, and hormonal changes. Instruments to measure satiety, quality of life, and cost effectiveness will also be administered. A detailed list of baseline measurements is shown in Table 3, and the methods used for measurements are shown in the ‘data collection’ section.

### 2.5. Randomisation

Following the baseline measurement, participants will be randomised into the MR group and control group at a 1:1 ratio. A third-party statistician will carry out the randomisation using the Winpepi computer program with minimisation method [19]. Gender (men, women), HbA1c (≤8.5% or >8.5%), and BMI (≤25 or >25) at the time of baseline measurement will be balanced in each group. Treatment allocation will not be blinded to both researchers and participants due to the nature of the intervention. The participants will be informed about which group they should accordingly be in before the intervention process starts.

### 2.6. Intervention

At baseline, both the MR and control group will receive diet consultation and printed diabetic education material from a research dietician. On top of that, the MR group will receive a meal replacement to replace one meal daily for 5 days a week. The duration of the intervention will be 12 weeks. All participants are required to return to the clinic in the 6th and 12th week for a post-intervention assessment. A final visit will be conducted in the 48th week for a sustainability assessment.

#### 2.6.1. Meal Replacement Arm (MR Group)

On top of dietary consultation, a diabetes-specific meal replacement will be provided without any charge for the participants in the MR group. The meal replacement we have chosen for this study is a nutritionally balanced liquid meal replacement product (Metabolic Sauver) specifically developed for patients with diabetes, which is formulated and produced by *Powerlife (M) Sdn Bhd,, Kuala Lumpur, Malaysia*. It is a low glycemic indexed (GI) drink with a GI of 26.84, which aims to improve the glycemic control of T2DM patients based on a combination of dietary fibre, slow-release carbohydrate isomaltose, and various plant extracts [18,20,21]. One of the main plant extracts in the meal replacement is cinnamon extract (*Cinnulin MS*), which has been shown to improve anthropometric parameters, glycemic control, and lipid profiles of T2DM patients [22,23]. A detailed nutrition sheet on Metabolic Sauver has been included in Appendix A, as has a table on the effects of some of the components in the formulation on obesity and T2DM.

Participants will be given instructions on how to prepare the meal replacement. A total of 75 g of meal replacement powder (as suggested by the manufacturer) will be dissolved in 300 mL of water to replace one meal of their choice (breakfast, lunch, or dinner). One serving of a meal replacement provides 327 kcal, 37.2 g carbohydrate, 16.2 g protein, and 14.4 g fat, and the participants are required to consume it for 5 days a week.

Participants are free to consume other meals at their will and continue their routine medical care without any changes. We believe that by enabling the participants to have flexibility in choosing their own mealtimes, their compliance with the meal replacement will be improved.

#### 2.6.2. Control Arm (DC Group)

Participants will receive one session of dietary consultation and education at baseline. The 90-min session will be conducted via a one-on-one basis by a research dietitian. The dietary consultation session mirrors usual/standard practice for diabetes patients at our local healthcare centres, including topics on calorie intake and carbohydrate requirements, carbohydrate food sources, carbohydrate exchange counting, sucrose intake, fat intake, Healthy Food Plate Malaysia, physical activity recommendation, and individualised menu plan.

Upon completion of the dietary consultation session, participants in the control group will be advised to continue their daily lives and routine medical care without extra intervention.

#### 2.6.3. Follow-Up

On the 6th and 12th week, all participants will return to the clinic for an outcome measures assessment. Upon completion of the intervention, participants will be monitored without any intervention for an additional 36 weeks. A final visit to the clinic will be conducted on the 48th week. During these last two visits, a similar assessment on the outcome measures will be carried out to determine the sustainability effect of meal replacement interventions.

### 2.7. Compliance and Adverse Effects Monitoring

A daily checklist will be handed out to the participants to monitor compliance with the meal replacement. Daily reminders will be sent through the WhatsApp application to remind the participants to ensure their compliance with the meal replacement. Non-responsive participants will be contacted through a phone call for an additional reminder. The participants in the treatment group will be required to bring along the meal replacement container during the follow-up visit to the clinic. Compliance will be assessed by comparing the daily checklist and the remaining meal replacement powder.

Participants will be considered non-compliant if they miss more than 30% of meal replacement intake throughout the intervention period. Participants will also be asked to report any complications or adverse effects, and this information will be recorded on an adverse effect sheet. Any occurrence of adverse events due to trial intervention will be reported to the clinic and attended to by a medical doctor from the research team if any treatment is needed.

### 2.8. Data Collection and Outcome Measurement

All participants are required to attend the clinic at baseline, 6th, 12th, 24th, 36th, and 48th week for data collection (Table 3).

#### 2.8.1. Socio-Demography Information

Participants will complete a demographic questionnaire during the screening process. The name, age, and gender of the participants will be recorded on their identification cards. Details such as ethnicity, phone number, address, marital status, number of children, educational status, years of formal education, occupation, and household income will be obtained through the questionnaire. The presence of pre-existing medical condition(s) and medical treatment(s) will be based on medical records presented by the participants and self-reports. Participants will also be asked about their habits on smoking and consumption of alcoholic beverages.

#### 2.8.2. Physical Activity Assessment

A validated Malay version of the Global Physical Activity Questionnaire [24] will be used to determine the physical activity level of the participants. The metabolic equivalent and total daily energy expenditure of the subject will be calculated based on the questionnaire’s guidelines.

#### 2.8.3. Dietary Assessment

An experienced research dietitian will be carrying out the dietary assessments on each participant at the beginning of the study. A 3-day dietary record will be used to monitor the dietary intake of the participants throughout the study period [25]. In the 3-day dietary record, food and drinks consumed by the participants for 2 weekdays and 1 day of the weekend in a week will be recorded. Participants will be asked to provide as much detail as possible on the meals taken, such as type and amount consumed, method of preparation, brands of food, sauce, and ingredients used. Participants will be shown pictures of various household measurements based on the Atlas of Food Exchanges & Portion Sizes to help them estimate the portion size [26]. Any supplements such as vitamins or minerals consumed will also be recorded. In addition, a specific modified validated food frequency questionnaire (FFQ) on added sugar food/drinks from Norimah AK [27] will also be used to identify the types, frequency, and quantity of added sugar to the food or drink consumption among participants. Computer-based analysis of mean caloric and nutrient intake will be performed using the Nutritionist Pro software (Axxya Systems, Stafford, TX, USA).

#### 2.8.4. Anthropometry Measurements

Anthropometric measurements will be conducted based on the procedures described by Gibson [28] and WHO [29]. We will be measuring weight, height, body composition (muscle mass, fat mass, and fat percentage), neck circumference, waist circumference, and hip circumference. Body mass index (BMI) will also be calculated.

Weight will be measured by using the Tanita HD-309 (Tanita Corporation, Tokyo, Japan) weighing scale. The standing height of the participants will be measured using a stadiometer to the nearest 0.1 cm, and BMI will then be calculated. Neck, waist, and hip circumference will be measured by using a soft measuring tape to the nearest 0.1 cm. The waist-hip ratio will be calculated by dividing the waist by the hip circumference.

Body composition will be measured by using bioelectrical impedance analysis with a body composition analyser (Model; SC-330, TANITA Japan) [30,31,32,33]. Muscle mass (kg), fat mass (kg), and fat percentage (%) will be extracted and copied into the database.

All anthropometry measurements will be conducted in duplicates, and mean readings will be calculated.

#### 2.8.5. Visual Analog Scale for Satiety Measurement

A Visual Analog Scale (VAS) questionnaire will be used to assess the participant’s perceived satiety after the consumption of the meal replacement in the treatment group or the consumption of a meal in the control group [34].

Participants will be instructed to fill up the VAS just before they consume a meal/meal replacement, 30 min, 60 min, 120 min, 180 min, and 240 min after meal consumption. An average appetite score will be calculated at each time of measurement by using the formula as shown below [35].

Average appetite = desire to eat + hunger + (10−fullness) + prospective food consumption.

#### 2.8.6. Blood Pressure

Ambulatory blood pressure will be measured using an electronic blood pressure machine (Omron Corporation, Kyoto, Japan).

#### 2.8.7. Metabolic Profiles

Blood samples of the participants will be collected by a qualified phlebotomist at baseline, on the 12th week and the 48th week. Prior to the day of blood collection, participants will be reminded to fast for eight hours (only plain water is permitted). After that, a total of 7.5 mL of venous blood sample will be taken for biochemical analysis. Blood will be transferred into their respective types of vacutainers and delivered to a certified diagnostic laboratory (Lablink (M) Sdn Bhd, Kuala Lumpur, Malaysia; MS-ISO 15189) for analysis. The results of the blood test will be mailed or given to the participants during the next follow-up. Contact details are provided along with the results to enable the participants to contact the investigators or physician if they have any inquiries. The biochemical parameters measured are listed below:

aHbA1c

Blood samples will be collected in an EDTA tube and measured using an automated high-performance liquid chromatography analyser Adams A1c HA-8180V (Arkray, Tokyo, Japan).

bFasting Blood Glucose

Blood samples will be collected in a sodium fluoride tube and measured using an automated photometric analyser ARCHITECT c16000 and c8000 (Abbott Laboratories, North Chicago, IL, USA).

cFasting Insulin Level

Blood samples will be collected in a serum separator tube, and the fasting insulin level is measured using an automated chemiluminescent analyser (Advia Centaur XP Immunoassay System, Siemens, USA). Insulin resistance index (HOMA-IR) is calculated according to the formula: fasting insulin (micro U/L) × fasting glucose (nmol/L)/22.5 [36].

dLipid Profile Test

Total cholesterol, HDL cholesterol, LDL cholesterol, and total glycerides will be analysed. Blood samples will be collected in a plain tube and measured using an automated photometric analyser ARCHITECT c16000 and c8000 (Abbott Laboratories, USA).

eLiver Profile Test

Total protein, albumin, bilirubin total, Alkaline Phosphatase (ALP), and Alanine Amino Transferase (AST) will be analysed. Blood samples will be collected in a plain tube and measured using an automated photometric analyser ARCHITECT c16000 and c8000 (Abbott Laboratories, USA).

fRenal Profile Test

Potassium, sodium, urea, creatinine will be analysed. Blood samples will be collected in a sodium fluoride tube and measured using an automated photometric analyser ARCHITECT c16000 and c8000 (Abbott Laboratories, USA). The estimated glomerular filtration rate will be calculated with the creatinine levels obtained.

#### 2.8.8. Hormone Level Measurements

Hormonal changes related to the glycemic control and satiety level of the participants will be measured in this study to understand the underlying pathway of meal replacement effects on obese and overweight T2DM patients. The hormones related to glycemic control that will be studied are adiponectin and leptin. The hormones related to satiety that will be studied are ghrelin, obestatin, and peptide YY. Adiponectin, leptin, ghrelin, obestatin, and peptide YY levels will be analysed through Enzyme-Linked Immunosorbent Assay (ELISA) commercial kits. The procedures for hormone analysing will be based on each kit protocol.

#### 2.8.9. Quality of Life

##### Audit of Dependent Diabetes Quality of Life (ADDQOL-19)

A validated Malay language translated ADDQOL-19 questionnaire will be used in this study to determine the quality of life (QoL) [32]. Good validity and reliability have been reported for ADDQOL-19 [32]. ADDQOL is comprised of 21 items with 19 items related to specific life domains and two overview items measuring general aspects [33]. The participants will be assisted by the researcher in understanding and completing the questionnaire.

##### Diabetes Treatment Satisfaction Questionnaire (DTSQ)

The Diabetes Treatment Satisfaction Questionnaire (DTSQ) will be used to measure participants’ satisfaction with the diabetes treatment regimens that include pre- and post-intervention [37]. DTSQ has been recommended as a useful tool by the World Health Organization (WHO) and the International Diabetes Foundation (IDF) for diabetes care outcome measurements [38]. It has been widely used to measure patient satisfaction with diabetes treatments.

Two versions of DTSQ will be utilised in this study, which are DTSQ status (DTSQs) and DTSQ change (DTSQc). DTSQs contain a six-item scale assessing treatment satisfaction and two items assessing the perceived incidence of hyperglycaemia and hypoglycemia. The six-item scale ranges from 0 (never) to 6 (always).

DTSQc contains the same eight-item scale but asks patients about their satisfaction with their current treatment as compared to their previous treatment [39]. It will be used to supplement DTSQs during post-treatment to overcome its ceiling effects where respondents who scored maximum or near-maximum satisfaction may show no improvement after the intervention [40]. The assessment of DTSQ will be performed by obtaining the mean scores and comparing them during pre- and post-intervention.

##### Diabetes Distress Scale (DDS)

The Diabetes Distress Scale (DDS) is a 17-item scale used to measure diabetes-related emotional distress [41]. A translated and validated Malay language version of DDS will be used in this study [42]. There are four critical dimensions of distress covered in DDS, including emotional burden (Item 1, 4, 7, 10, 14), regimen distress (Item 6, 8, 3, 12, 16), interpersonal distress (Item 9, 13, 17), and physician distress (Item 2, 5, 11, 15).

#### 2.8.10. Cost-Effectiveness

Quality-adjusted life years (QALY) will be the main outcome parameter for the evaluation of cost-effectiveness from a societal perspective. EuroQol five-dimension Inventory (EQ-5D-5L) will be utilised in this study to calculate QALY. It is the most used instrument worldwide in economic evaluations that measure health benefits in terms of QALY [43]. English and Malay versions of EQ-5D-5L, which have been validated for use in the Malaysian population, will be utilised in this study [44,45].

EQ-5D-5L is a self-report questionnaire that assesses five life domains, including mobility, self-care, usual activities, pain/discomfort, and anxiety/depression [43]. There is also an EQ-VAS section to measure respondents’ perception of their overall current health on a vertical visual analogue scale with ‘100′ corresponding to ‘The best health you can imagine’ and ‘0′ representing ‘The worst health you can imagine’.

Scores on the five dimensions will be totalled up and converted into utility weights for computing QALY using the EQ-5D algorithm. Analysis will be conducted to estimate the direct and indirect costs for the intervention group and the control group for routine diabetes care. Intervention costs will then be estimated by using a bottom-up micro-costing analysis that will include meal replacement cost and operational costs such as dietician’s salary. The cost for control group or routine diabetes care will be estimated from a questionnaire on participant’s health care utilisation (general practitioner, dietician, physiotherapist, consultations at outpatient clinic, and hospitalisation) and use of medication.

The cost per QALY gained will be computed, and incremental cost-effectiveness ratios (ICER) will be calculated as the difference in costs divided by the difference in QALYs between the intervention and the control group using a bootstrap analysis. A cost-effectiveness plane can be plotted to indicate which intervention is more effective or expensive than the other.

### 2.9. Statistical Analysis

Statistical Package for the Social Sciences (SPSS) program version 23.0 will be used to analyse the data. The data will be presented as mean ± SD for parametric data and median (range) for non-parametric data. The normality of data distribution will be tested using the Shapiro–Wilk test. A *p*-value of <0.05 will be used to denote statistical significance. Descriptive analysis including frequencies, percentage, mean, and standard deviation will be used to analyse sociodemographic data and blood pressure.

For the effects of meal replacement on metabolic profiles, anthropometric measurements, satiety levels, hormonal changes, and QoL scores, mixed-design ANOVA will be used to compare between the intervention group and the control group to determine the interaction effect of intervention and time.

To determine the cost-effectiveness, incremental cost-effectiveness ratios (ICER) will be calculated by dividing the difference in total costs (between control and treatment) with the difference in the chosen measure of health outcome or effect (QALY). To assess the sustainability of the meal replacement, mixed-design ANOVA will be used to compare clinical parameters and hormonal level changes between groups on week 12 and week 48.

The main analysis will be conducted in the intention-to-treat (ITT) population in which missing values will be predicted by using the estimating equation method by adapting statistical models to the data observed using the missing-at-random method. Complete case analysis will also be performed in the per-protocol (PP) population.

### 2.10. Patient and Public Involvement

Neither the patient nor the public is involved in the conception, design, or execution of the study. However, participants will be informed about the outcome of the trial via one-to-one consultation and email upon trial completion.

### 2.11. Data Monitoring

The research team will monitor the data internally and have weekly meetings to discuss the progress of the trial and data collection. There is no external data monitoring board. Each participant will be assigned a unique identifier code to be entered in the database. Only patient codes will be entered into the database to ensure the anonymity of the participants. The data will only be entered and accessed by members of the research team. The database will be password protected and encrypted. Missing data will be identified and mentioned with reason. Data obtained throughout the trial will be stored for a duration of 15 years. Handling of personal data will be carried out in compliance with the Malaysia Personal Data Protection Act (PDPA) 2010.

### 2.12. Ethics and Dissemination

This study has obtained human ethics approval from the Human Research Ethics Committee UKM (JEP-2019-566). We have also registered this study protocol at the Thai Clinical Trials Registry (Medical Research Foundation, Bangkok, Thailand; TCTR ID: TCTR20210921004). Any amendments to the protocol will be submitted to the ethics committee for review and will be conveyed to all relevant parties. Written consent (Online Appendix A) will be obtained from all participants before the commencement of the study. Each participant will be assigned a unique identifier to be used in the database. The personal identity of all participants will not be used for any public purpose or for publication, nor will it be accessible to personnel outside of the research team. Results from this study will be disseminated via peer-reviewed publications and presented at relevant international conferences.

## 3. Impact of COVID 19 on Data Collection

### Preliminary Result

The study is currently in the recruitment and initial intervention phase. A total of 5185 patients were screened between September 2020 to May 2021 from digital medical records. A total of 136 participants were identified to be recruited. The intervention process has not yet been started. The overall distribution of participants identified by gender, age, and race is shown in Table 4 below. The overall mean age (±SD) of the participants is 54.9 ± 9.3 years. Among them, there are 81 males and 55 females with predominantly 63.97% Malay, 18.38% Chinese, 15.44% Indian, and 0.02% others. Overall, HbA1c levels of the participants averaged 8.5 ± 1.0%, while fasting plasma glucose levels averaged 8.6 ± 2.5 mmol/L.

## 4. Discussion

The global epidemic of obesity and associated T2DM direly needs a greater emphasis on effective lifestyle intervention programs for patients. Meal replacement is often included as a part of the lifestyle intervention program as it is effective as an initial phase for diabetic patients to lose weight in a therapeutic program [46]. Previous trials utilising meal replacements on obese or T2DM patients have demonstrated significant improvements in HbA1c, weight, lipid profile, insulin sensitivity, and cardiovascular risk factors [47,48,49,50,51].

This prospective, randomised, controlled open-label trial will focus on evaluating the effectiveness of a diabetes-specific meal replacement formulation on the weight loss and glycemic control of obese and overweight T2DM patients. To date, this is the first trial of its kind to test the efficacy of this low glycemic indexed meal replacement formulation containing cinnamon extract (*Cinnulin MS*) on overweight or obese T2DM patients. There is also a lack of studies on short-term meal replacement intervention in a sufficiently powered and well-designed clinical trial in Malaysia. We hypothesised that the meal replacement intervention would be effective in maintaining weight loss, improving glycaemic control parameters, and improving the quality of life of the participants.

Apart from anthropometry and biochemical analysis, a cost-effectiveness analysis of the meal replacement intervention will also be conducted, in comparison with the standard care in public. It is anticipated that this meal replacement intervention in T2DM management would be more cost-effective, as simulated in the other study (Stephen 2010).

Meal replacements have been proven to improve patients’ weight and glycaemic control. However, due to the lack of further intervention following the completion of the initial intervention, it has also been shown to be less effective in maintaining glycaemic control in some meal replacement studies. This is shown in a study by Ash et al., in which the HbA1c levels of the participants who received a 12-week diet intervention with meal replacement returned to baseline levels after their 18th month follow-up [16]. Similar results of weight and HbA1c levels rebounding have been reported in other studies [52,53,54].

The sustainability of the meal replacement formulation to maintain the effect of weight reduction and glycemic control will be assessed by the end of this study through constant monitoring of the participants for another 36 weeks. Possible adverse effects among the participants in the meal replacement trial that we anticipate may occur are nausea or hypoglycemia. The participants will be advised to report any episodes of adverse effects directly to the medical doctor in the research team immediately.

Nevertheless, the commencement of this clinical trial has met with unprecedented challenges due to the COVID-19 pandemic. Most resources and policies have shifted to focus on the management and prevention of the pandemic in the country. As a result, our research progress has been confronted with various restrictions and delays. COVID-19 has led to a restriction of patient visits to clinics for non-essential purposes. Hence, the recruited participants for our study are not permitted to visit the clinic for baseline measurement and intervention.

The European Medicines Association and the US Food and Drug Administration have come out with a series of guidelines and suggestions on conducting clinical trials during the pandemic for better patient protection and to enable continuous trial execution without compromising good clinical practice standards [55,56]. Doherty et al. [57] have reported that one important consensus in handling clinical trials during COVID-19 is reducing the potential COVID-19 exposure risk to patients by minimising hospital visits. Dorsey et al. [57] have suggested that COVID-19 has led to the rise of decentralised studies in which studies may be designed as site-agonistic and flexible in location. Taking the guidelines and approaches from other trials as a reference, we have proposed and outlined a hybrid method for an intervention session that comprises both virtual and physical sessions to mitigate the impact of the delay in the intervention progress [55,56,57,58,59].

One of these methods is to utilise telecommunication software to conduct the diet education online for digitally literate participants. Based on a survey by the American Society of Clinical Oncology on clinical trial participants, nearly all of their respondents (90.3%) indicated telehealth as a potential alternative for conducting clinical trials during the pandemic [60]. Step-by step instructions will be provided to the participants on utilising the online meeting software for conducting diet education. Patient reviews of compliance to meal replacement and adverse events can also be monitored through telecommunications methods such as email, phone, or messaging applications. A study by Aravindhan et al. [61] has reported that virtual interviews for older research participants, because of lockdown measures, are feasible in the current pandemic situation to minimise physical sessions.

Home visits will also be carried out for patients who are unwilling to attend the site for intervention. The phlebotomist and a researcher will visit the patient’s house for blood taking, assisting in the completion of questionnaires and providing instructions on the trial. Strict standard operating procedures will be adhered to in order to ensure the safety of the trial participants and the researchers from COVID-19. This is as suggested by the FDA, who states that consideration should be given for alternative locations for visitations if a patient is not able to go to the investigational site [55]. The investigational product, which is the meal replacement powder, will also be delivered to the participants’ homes [62].

Another possible challenge that we expect to face in this study is maintaining the compliance of the participants who undergo the meal replacement intervention. Although there is no consensus on the acceptable minimum adherence level in clinical trials, it is important to maintain a high level of adherence, as low adherence can lead to a negative impact on the result interpretation and statistical power [63,64]. Mogre et al. reported a middle diet adherence rate of 58.1% for diabetes patients in middle- and low-income countries [65]. An adherence rate between 43% and 78% of participants receiving treatment during a clinical trial for chronic conditions can be classified as being compliant [66]. To minimise the risk of a high dropout rate, the participants will be monitored closely via phone contact, video call, and Whatsapp messages. Daily reminders will be sent to remind the participants of adhering to the meal replacement routine. Participants will be given flexibility in choosing which days they prefer to consume the meal replacement on as long as they choose 5 days a week.

### 4.1. Limitation

The major limitation of this trial is that it is a single-centred study. Hence, the generalisability of the study might be limited to the Malaysian T2DM population who visit the public health sector to seek treatment. Meal replacement products are also given to the participants without any charge. This might also impact the generalisability of the study, as dropout rates for less motivated patients in real-world settings in such intervention programs are likely to be higher. To reduce this limitation from the generalisability perspective, we have used randomisation techniques during sampling to maximise the representation of our target population.

Due to the nature of the study, the treatment allocation will not be blinded to both participants and the researchers. However, to minimise selection bias, randomisation into groups and statistical analysis will be carried out by a third-party statistician.

The taste and texture of the meal replacement we are studying might also affect the compliance of the participants, as there are reports from testers who dislike the cinnamon taste of the liquid meal replacement. We will inform our study participants beforehand to get them to gradually adapt to the taste.

### 4.2. Implication and Future Implementation

The recruitment: intervention and follow-up process will have approximately a 2-year delay as a result of the COVID-19 pandemic. We expect the results to be available and publishable by June 2023. This study will enable us to assess the potential of this low glycaemic indexed meal replacement and will allow for it to be included as a part of lifestyle intervention and implemented into present healthcare programs. This research can also help in assessing the sustainability of meal replacements upon the secession of the intervention. Future research can be carried out to develop a lifestyle intervention module incorporating a short-term meal replacement therapy for the maintenance of T2DM patients.

## Figures and Tables

**Figure 1 ijerph-19-04188-f001:**
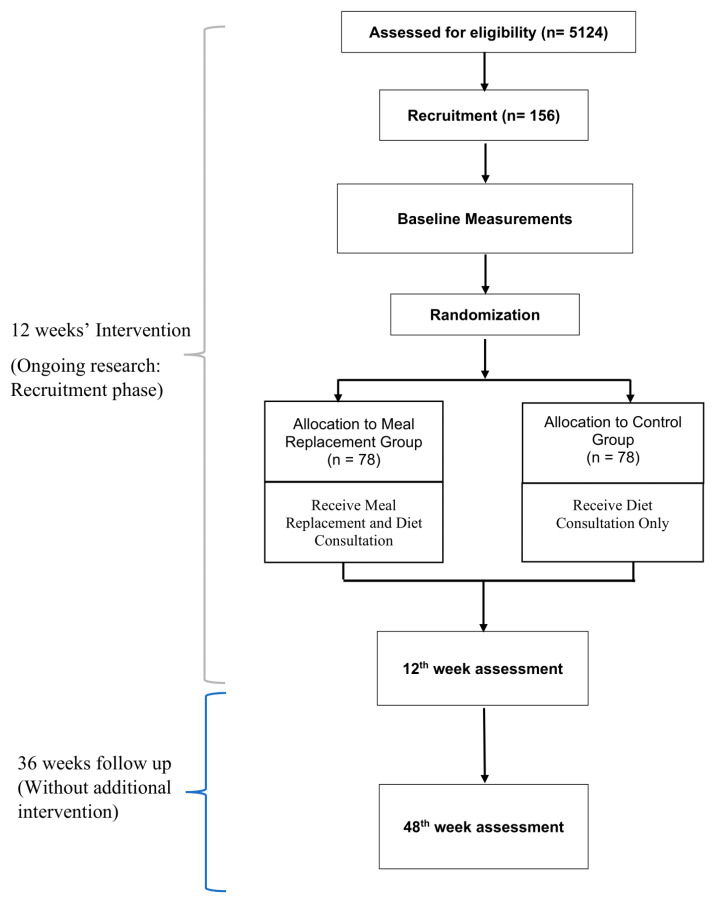
Research Flow Chart.

**Table 1 ijerph-19-04188-t001:** Inclusion and Exclusion Criteria.

Inclusion Criteria	Exclusion Criteria
Aged 20–65 years old	On insulin treatment
Diagnosed with T2DM for at least 6 months with baseline HbA1c levels between 7.5% and 12% for the past 3 months	With chronic kidney disease or on continuous ambulatory peritoneal dialysis or hemodialysis (GFR < 30 mL/min/1.73 m^2^)
Overweight or obese with BMI ≥ 25 kg/m^2^	With hepatic diseases (ALT > 120 IU/L)
On stable doses of any oral hypoglycaemic agents for the past 3 months	With history of chronic alcohol abuse
	Pregnant and lactating women
	Currently consuming any weight reduction products or any slimming prescriptions
	Currently involving in weight loss programs
	Record of COVID-19 diagnosis

**Table 2 ijerph-19-04188-t002:** Protocol for calculation in G*Power.

Sample Size Calculation	
F tests—ANOVA: Repeated measures, between factors	
Analysis: A priori: Compute required sample size	
**Input**	
Effect size f	= 0.2055067
α err prob	= 0.05
Power (1-β err prob)	= 0.80
Number of groups	= 2
Number of measurements	= 4
Corr among rep measures	= 0.5
**Output**	
Noncentrality parameter λ	= 8.1087367
Critical F	= 3.9214782
Numerator df	= 1.0000000
Denominator df	= 118
Total sample size	= 120
Actual power	= 0.8063091

**Table 3 ijerph-19-04188-t003:** Data collection Table.

Data Collected/Session	Week 0	Week 6	Week 12	Week 24	Week 36	Week 48
Socio-demography information	/					
Global Physical Activity Questionnaire	/		/	/	/	/
Dietary Assessment						
3-day dietary recall	/	/	/	/	/	/
Food Frequency Questionnaire	/	/	/	/	/	/
Anthropometry Measurements						
Height	/	/	/	/	/	/
Weight	/	/	/	/	/	/
Neck circumference	/	/	/	/	/	/
Waist circumference	/	/	/	/	/	/
Hip circumference	/	/	/	/	/	/
Body composition (muscle mass, fat mass, fat percentage)	/	/	/	/	/	/
Metabolic Profiles						
HbA1c	/		/			/
Fasting Blood Glucose	/		/			/
Insulin Resistance Index (HOMA-IR)	/		/			/
Lipid profile	/		/			/
Renal Profile	/		/			/
Liver Profile	/		/			/
Hormone related to Glycemic Control						
Adiponectin, Leptin	/		/			/
Satiety Measurements						
VAS	/	/	/	/	/	/
Ghrelin, obestatin, peptide YY	/		/		/	/
Quality of life						
ADDQOL-19	/		/	/	/	/
DTSQ	/		/	/	/	/
DDS	/		/	/	/	/
Cost-effectiveness						
EQ-5D-5L	/		/	/	/	/

**Table 4 ijerph-19-04188-t004:** Sociodemographic of participants identified (n = 136).

Parameters	Numbers/Mean
Gender	
Male	81 (59.5%)
Female	55 (40.5%)
Age	54.9 ± 9.3
Race	
Malay	87 (63.97%)
Chinese	25 (18.38%)
Indian	21 (15.44%)
Others	3 (0.02%)
Duration of diagnosis for T2DM (Years)	6.4 ± 4.3
HbA1c Levels (%)	8.5 ± 1.0
Fasting Plasma Glucose (mmol/L)	8.6 ± 2.5

## Data Availability

The data presented in this study are available on request from the corresponding author.

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
