# Peer review of "Efficacy and Sustainability of Diabetes-Specific Meal Replacement on Obese and Overweight Type-2 Diabetes Mellitus Patients: Study Approaches for a Randomised Controlled Trial and Impact of COVID-19 on Trial Progress"

_ijerph, 2022, doi:10.3390/ijerph19074188_

Round 1

Reviewer 1 Report

The subject is interesting and has an impact on public health.

However, the manuscript is very confusing, it is not clear that a protocol is proposed to follow in future studies since it is presented as a proposal for a future study. Authors should review protocol articles or study guides and rewrite and order their work accordingly.

 The authors present a proposed protocol to be used to conduct a future study of public health importance, however it is presented as materials and methods for a future study, not as a proposed protocol to measure the efficacy of meal replacement in overweight T2DM.

In the abstract it should be clear that this work is a proposal for a protocol or guide for future studies and in its content it should be formulated as a guide, not a study proposal to be carried out.
The lines of the document are not numbered and it makes it difficult to review.

Tittle

T2DM First time mentioned, add meaning

Abstract

HbA1c First time mentioned, add meaning

This is part of materials and methods:thics and dissemination: This study has obtained human ethics approval from RECUKM (JEP-2019-566) and registered at Thai Clinical Trials Registry (TCTR ID: TCTR20210921004).

Keywords: You need at least six words

The lines of the document are not numbered and it makes it difficult to review

Page 2: First time mentioned, add meaning

Page 3 paragraph 3:

Study or studies proposed to be conducted with the proposed protocol?

Author Response

25th February 2022

Dear Editors and Reviewers,

We are truly grateful on your kind consideration of our manuscript for publication in your reputable journal. We would also like to thank the reviewers for reviewing through our text and providing constructive suggestions for us to improve the quality of this manuscript. 

All comments are being considered carefully and amended accordingly to improve the quality of the manuscript. In below we have attached a table of response for respective reviewer’s comments. We believe that the helpful suggestions from reviewers have improved the overall strength of our manuscript, both in the methodology section and discussion section. We have also sent the manuscript for detailed proofreading from a professional third-party proofreader to improve structure, grammar, and spelling of the text.

We have attached the revised manuscript in the submission system where all the amendment based on reviewers’ comments is tracked with track changes.

We hope that the corrections on the manuscript based on suggestions from the editor and reviewers would facilitate the decision to publish the study in your journal. We are also open to any future comments or suggestions for further improvements on our manuscript.

Look forward to your valuable decision. Thank you for your support.

Sincerely,

Dr. (Ph.D) ARIMI FITRI MAT LUDIN

Center for Healthy Ageing and Wellness, H-CARE

Faculty of Health Sciences

Universiti Kebangsaan Malaysia

50300 Jln Raja Muda Abd Aziz

Kuala Lumpur MALAYSIA.

[email protected]

+603-9289 8043 (Office)

Reviewer 2 Report

This a study protocol describing a prospective, randomized, controlled open label trial aiming at evaluating the effectiveness of a diabetes-specific meal replacement formulation on weight loss and glycemic control of obese and overweight T2DM patients. As such no results are presented and therefore, I don't know if there is the possibility of publishing a study protocol without specific results. 

The study protocol is well described and organized.

Some aspects could be improved.

The introduction should be shortened.

The part relating to the impact of Covid-19 on patient recruitment is well known and could be reduced to a sentence or two (the Authors dedicate to much space to this issue).

Meal replacement features are very few and should be more detailed; also the composition and the rational for its composition should be highlighted.

Apart from these findings, I return to the main point; I have no idea if a study protocol can be published.

Author Response

(The authors gave the same response as above.)
